# Morphology of *Penicillium rubens* Biofilms Formed in Space

**DOI:** 10.3390/life13041001

**Published:** 2023-04-13

**Authors:** Megan Hupka, Raj Kedia, Rylee Schauer, Brooke Shepard, María Granados-Presa, Mark Vande Hei, Pamela Flores, Luis Zea

**Affiliations:** 1Molecular, Cellular, and Developmental Biology Department, University of Colorado Boulder, Boulder, CO 80309, USA; 2Smead Aerospace Engineering Sciences Department, University of Colorado, Boulder, CO 80309, USA; 3BioServe Space Technologies, Aerospace Engineering Sciences Department, University of Colorado, Boulder, CO 80309, USA; 4Universidad del Valle de Guatemala, Guatemala City 01015, Guatemala; 5Johnson Space Center, NASA, Houston, TX 77058, USA

**Keywords:** microgravity, *Penicillium chrysogenum*, fungal biofilm, International Space Station, microscopy, spacecraft

## Abstract

Fungi biofilms have been found growing on spacecraft surfaces such as windows, piping, cables, etc. The contamination of these surfaces with fungi, although undesirable, is highly difficult to avoid. While several biofilm forming species, including *Penicillium rubens*, have been identified in spacecraft, the effect of microgravity on fungal biofilm formation is unknown. This study sent seven material surfaces (Stainless Steel 316, Aluminum Alloy, Titanium Alloy, Carbon Fiber, Quartz, Silicone, and Nanograss) inoculated with spores of *P. rubens* to the International Space Station and allowed biofilms to form for 10, 15, and 20 days to understand the effects of microgravity on biofilm morphology and growth. In general, microgravity did not induce changes in the shape of biofilms, nor did it affect growth in terms of biomass, thickness, and surface area coverage. However, microgravity increased or decreased biofilm formation in some cases, and this was incubation-time- and material-dependent. Nanograss was the material with significantly less biofilm formation, both in microgravity and on Earth, and it could potentially be interfering with hyphal adhesion and/or spore germination. Additionally, a decrease in biofilm formation at 20 days, potentially due to nutrient depletion, was seen in some space and Earth samples and was material-dependent.

## 1. Introduction

Biofilms are cell populations that grow embedded in an extracellular matrix (ECM), which has effects on its ability to adhere to itself and to surfaces, and which changes the interaction between cells and nutrients, quorum-sensing molecules, and its environment in general [1]. For example, it is estimated that up to 80% of bacterial and archaeal cells in nature grow as biofilm populations [2]. The ability of fungi, eukaryotic organisms, to form biofilms depends, in part, on the substratum upon which they form [3], and occurs in a six-phase fashion: arrival at substratum, adhesion, colonization, ECM production, biofilm maturation, and dispersal [4].

One of the most important phases in biofilm formation is adhesion and depends largely on cell–surface and cell–cell adhesion. Following adhesion, microbes proliferate and form the ECM. The ECM has antimicrobial-resistant properties complicating the removal of fungal cells from surfaces. If the process of adhesion is blocked, proliferation and formation of the extracellular matrix will be prevented, providing an ideal way to stop biofilm development [5]. The factors contributing to the fungal biofilm adherence are dependent on the species and strain. Past studies have identified hydrophobicity as a determinant for fungal biofilm adhesion, while other studies concluded that hydrophobicity of the material alone is not enough to determine adhesion [6,7]. The lack of a consensus on the role of material hydrophobicity in biofilm adherence complicates the development of a biofilm-resistant material.

Because fungal biofilms are challenging to remove from surfaces due to the ECM, fungal biofilms, or molds, which in the context of this investigation are considered equivalent to fungal biofilms, are a concern for human spaceflight for at least two reasons. The first comes from the potential damage to the surfaces (namely equipment) upon which they grow. Fungal biofilms can directly damage surfaces by using the material as a source of energy, or indirectly by degrading it with enzymes and other metabolic byproducts [8]. This is not an unfounded concern, as there have been issues derived from fungal biofilm formation in previous space stations. On the Soviet Salyut 6 space station, fungal biofilms were observed on “piping and equipment behind the panels” [8]. On Salyut 7, fungal growth was found on the water condensation system used for water recycling, the radiator of the thermal system, and on electric connectors and cables, in some of which “holes” on the insulation tape were observed [8]. On the Mir space station, fungal biofilms (*Penicillium chrysogenum* now called *Penicillium rubens* [9]) started growing on windows’ rubber seals and expanded onto (and damaged) the quartz and titanium frame. Fungal biofilms were also reported on components of the air conditioning system and on cables using the electric insulation as a source of food. Furthermore, there were “zones completely covered with growth of mold fungi destruction of the plates” on the oxygen electrolysis block, and fungal biofilms were observed on the head phone of the EVA suit, lavatory pan, piping of the thermal control system, urine processing assembly, etc. Cosmonauts reported fungal growths, usually observed on moist areas where water condensed over “cold” material surfaces, as yellow or orange, or as “white powder”, sometimes “the size of a football” [8]. Both in the Mir space station as well as in the International Space Station (ISS), *Penicillium* has been the most isolated genus of fungi [10]. Many of the fungi isolated from Mir were known biodegraders of polymers and showed more aggressive post-flight colony formation and biodegradation activity than reference strains, which the authors suggested could have been caused by the exposure to changes in radiation level, magnetic fields, or solar activity [11].

The second reason why fungal biofilms are a concern to human spaceflight is medical in nature. The ECM and altered cell–cell communication via quorum-sensing molecules can increase antifungal resistance and pathogenicity compared to planktonic cells [1,12], and the biofilms can serve as reservoirs of persistent sources of infections [3]. Fungi can cause infections in humans that are harder to diagnose and treat than those of bacterial origin [3], resulting in failure of antifungal therapy, which may require surgical intervention [4] and translate into high mortality rates [3]. Catheters can be colonized by fungal biofilms, from which cells can detach and cause acute fungemia and disseminated infection that may require long-term antifungal treatments. Similarly, fungi can infect the respiratory tract, causing bronchitis and other types of infections [4]. Fungi can cause superficial and invasive infections, [13], particularly among the immunocompromised [14], posing a threat to astronaut health as spaceflight has been shown to compromise immune response [15] and confined environments, such as spacecraft, can increase exposure [16,17]. Particularly, *Penicillium rubens* has been reported to cause rare but severe cases of esophagitis, endophthalmitis, and invasive pulmonary mycosis among immunocompromised individuals [18,19,20].

While these challenges are significant for space stations orbiting Earth, even greater complications can arise in long-term missions. For example, in case of an emergency, such as critical equipment failure or urgent medical needs, the crew can potentially be back on the ground in a matter of hours. However, this will not be true for missions beyond the lower Earth orbit (LEO), including trips around the Moon, Earth–Mars transit, or on space habitats on the Moon and Mars [21]. Although microbial planktonic growth has been studied in relative detail in space [22,23,24,25], little is known about fungal behavior in space when grown as biofilms. In fact, to the best of our knowledge, fungal biofilm growth in space has not been assessed and compared to ground controls, particularly how surface area coverage, biomass, and biofilm thickness change as a function of gravity, time, and material upon which they grow. Conversely, focus has been given to bacterial studies [26,27,28]. This was one of the objectives of our Space Biofilms project performed on the ISS. Here, we report on morphological characteristics of *P. rubens* biofilms grown for 10, 15, or 20 days on seven different spacecraft- and nosocomial-relevant materials on board the ISS, and how they compare against their matched ground controls. Separately, we have reported on the original design of this experiment [9] and on the changes in bacterial biofilm formation in microgravity, also as part of our Space Biofilm project [29,30,31]. Due to the frequency of fungal contamination on spaceflight hardware and habitats and the risks fungal colonization poses to astronaut health and mission success, it is critical to better understand the effect of spaceflight on fungal biofilm development.

## 2. Materials and Methods

### 2.1. Test Matrix: Fungal Strain, Materials, and Experimental Organization

*Penicillium rubens ATCC*®* 28089™* was chosen given the genus ubiquity in space stations and this specific strain’s presence (and damage caused) in the Mir space station [8,11]. The experiment examined the effect of (i) gravity (microgravity on board ISS vs. 1 g control in our lab on Earth), (ii) incubation duration (10, 15, and 20 days), and (iii) material substratum (seven materials tested) on fungal biofilm morphology. As described in further detail in [9], the materials chosen (and why there were chosen) for this experiment were:Aluminum Alloy (Al6061), as it is used in spacecraft structures, thermal control, structures for electronic devices and panels, etc.;Stainless Steel 316 (SS316), as it is used in spacecraft environmental control and life support system (ECLSS) tanks and tubing (including for potable water), and Extravehicular Mobility Unit (EMU) elements, and on Earth in surgical equipment and implants;Quartz, as it is used in spacecraft windows (at least one of them in the Mir space station damaged by *P. rubens*);Silicone, as it used in space in O-rings and on Earth in urological catheters, surgical incision drains, and respiratory devices;Titanium alloy (Ti-6Al-4V), as it is used in spacecraft structures, antennae, pressure vessels, brackets, fittings, etc., and on Earth in implants;Carbon fiber, as it is used in spacecraft aeroshells and other applications;MIT Nanograss, chosen to be interrogated as a potential solution to fungal biofilm formation in space. This was developed by the Massachusetts Institute of Technology (MIT) and is the substrate (nanoetched silicon wafer) described in the making of a lubricant-impregnated surface (LIS) without the lubricant oil and treated to increase hydrophobicity [32].

For each of these materials except SS316, six replicates of each incubation time were launched to ISS to be fixed with 4% paraformaldehyde (PFA) at the end of the incubation period (for the post-flight morphology studies presented in this manuscript). SS316 was set up similarly but had 12 replicates per incubation time to enable more in-depth morphology analyses. Additionally, all materials had seven replicates of per incubation time launched to the ISS to be preserved in RNAlater (except MIT’s Nanograss, which only had 6 replicates) for post-flight differential gene expression analyses (samples available through NASA’s space Biology Biospecimen Sharing Program). This yielded a total of 288 space samples. An equivalent set (288 samples) was prepared at the same time as the Earth control. Additionally, a time zero (*t*_0_) set of samples that included seven replicates per material was also prepared.

### 2.2. Pre-Launch Preparation

The following describes the preparation of the Space Biofilms fungal samples prior to launch. Preparation of space and Earth control samples took place simultaneously at the Eastern Virginia Medical School (EVMS), unless otherwise specified.

#### 2.2.1. Fungal Strain

One lyophilized vial of *P. rubens* was rehydrated with sterile water for 30 h, and then used to inoculate potato glucose agar (PGA, Sigma-Aldrich, St. Louis, MO, USA, Cat. 70139) plates. After incubation for eight days at 25 °C, the plates were flooded with 6 mL of 1X PBS (Sigma Cat. P4417) and colonies were gently rubbed with an inoculation loop to dislodge spores. The spore solution was then used to inoculate new PGA plates (working plates) in a 6 × 6 grid fashion. The working plates were six days old when used to inoculate the material coupons with spores (Section 2.2.5).

#### 2.2.2. Preparation of Syringes

To provide a humid environment ideal for fungal growth, a couple of absorbent mats inside the flight hardware needed to be wet at the experiment’s start (once in microgravity) (described in more detail in Section 2.2.5). Additionally, the experiment’s termination for morphology samples consisted of fixation with 4% PFA, while for transcriptomic samples, of preservation in RNAlater. In preparation for these tasks, 3 mL syringes filled with 3 mL of sterile distilled water, and 5 mL syringes filled with 4 mL of 4% PFA (diluted with 1X PBS from 16% PFA, Alfa Aesar, Haverhill, MA, USA, Cat. 43368) or RNAlater (Sigma-Aldrich, Cat. R0901) were prepared ahead of time.

#### 2.2.3. Preparation of Risers

Silicone risers (Saint Gobain, Courbevoie, France, Cat. D1069809) were shortened to 13 mm height at BioServe Space Technologies, bag washed with 1% Liquinox (m/v) solution (Alconox, White Plains, NY, USA, Cat. No. 16-000-126) in distilled water, rinsed with distilled water, dried at 100 °C, and autoclaved for 30 min at 121 °C (dry cycle) before plate assembly (Section 2.2.5). More details on why these were used and how they were implemented are given in Section 2.2.5.

#### 2.2.4. Preparation of Material Coupons

The 1 cm^2^ material coupons (Table 1) were labeled and organized by plate ahead of time at BioServe Space Technologies (except Silicone, which was labeled after cleaning). When at EVMS, the labeled coupons were washed with 1% Liquinox by ultrasonic cleaning, rinsed with distilled water, and then dried at 100 °C. Silicone coupons were cleaned by bag wash, rinsed, dried, and subsequently labeled. MIT Nanograss coupons were not cleaned, as they were prepared in sterile conditions.

All coupons were submerged in 70% ethanol for 15 min on each side and left to air dry inside the biosafety cabinet to disinfect the material surfaces. Once completely dried, the coupons were submerged in sterile Potato Dextrose Broth (Sigma-Aldrich, P6685) for 30 min and allowed to air dry overnight at room temperature; this supplemented the surface with nutrients to promote fungal growth over the coupons. The next day, coupons were carefully handled (never touching the surface, only the edges) to stick the cardboard side of 1 cm^2^ pieces of double-sided tape (3M, Saint Paul, MN, USA, Cat. No. 9731) to the bottom of the coupons (over the label). The clear side of the tape was then peeled in preparation for spore inoculation (Section 2.2.5).

#### 2.2.5. Inoculation of Spores and Sample Assembly into Hardware

Each coupon was inoculated with fungal spores per the “dry conidia” method [33]. A 5 g weight (McMaster, Hamilton, ON, Canada, Cat. 1777T23) was stuck to the back of the coupon using double-sided tape and then gently placed face down on top of 6-day-old mold of *P. rubens* grown on PGA plates (Figure 1a,b). Each coupon was placed on a new unused area of the mold for exactly 10 s to transfer comparable amounts of spores (on average 777 spores/mm^2^, Appendix A) onto each coupon’s surface. 

One sterile riser was inserted in each well of a commercial-off-the-shelf (COTS) 24-well plate (CellTreat, Pepperell, MA, USA, Cat. 229524) (Figure 2a). An inoculated coupon was then placed on top and adhered to the riser (Figure 2b) and secured in place by pressing gently on the corners. The plates contained 24 coupons placed in a random fashion (ISS and Earth equivalent plates matching) to reduce the effect of a possible humidity or oxygen gradient in the Plate Habitat (PHAB). To ensure the safety of the crew and the space station, a sterile and gas-permeable Breathe-Easy membrane (Sigma, Cat. Z380059) was placed over each plate (Figure 2c) to seal and contain the fungi and spores. The use of the silicone risers reduced the working volume in the well to 0.7 mL, rising the material coupon closer to the membrane, and minimizing the volume of fixative/preservative needed to terminate the experiment. This resulted in reduced crew time, launch mass, and volume.

Four loaded plates (inside a bag) and a temperature and humidity recorder were housed in each of the spaceflight hardware: BioServe Space Technologies’ PHAB (Figure 3). The temperature and relative humidity (RH) recorder was used to determine whether the environment was similar between PHABs. The PHAB was used as the secondary containment for the samples as well as a tool to maintain humidity and promote fungal growth. Each PHAB was designated to a particular incubation time and gravitational condition; three were sent to the ISS (one corresponding to 10, 15, and 20 days of incubation), while three equivalent PHABs were kept on Earth as controls.

The assembled PHABs were transferred to cold stow (4 °C), where they remained at this temperature for pre-launch, launch, and ISS stowage until experiment activation. The reduction in temperature and the placement of plates inside a plastic bag to reduce gas permeability help prevent activation (fungal growth) until samples reach microgravity. Additionally, the PHABs were maintained oriented during pre-launch stowage and during launch so that the coupons were on top to minimize the risk of detachment from the risers.

#### 2.2.6. Sample Set *t*_0_

The *t*_0_ samples did not undergo any cold stowage, but were immediately activated after inoculation by transferring to 25 °C for 6 h to allow spores to adhere to the surface, then fixed with 4% PFA on Earth. The objective of including these samples was to determine if they showed a discrepancy in spore count between materials at the start of the experiment and to determine if later comparison across material type was valid, or a result of unequal inoculation. The 6-hour incubation was chosen to allow enough time for spore adherence to prevent spores from washing away during staining.

### 2.3. Experiment Performance—On ISS and on Earth

The Earth controls were performed following the same timeline as the flight set, albeit with a 2-hour delayed start, described as step 3 in Figure 4 (asynchronous). This allowed the spaceflight session to conclude prior to start the equivalent on Earth. Therefore, the operations described in this section (Figure 4) pertain to both space and Earth control sets.

The start of the experiment began at sample “activation” when samples were removed from cold stow (after ~12.5 days) and provided with high relative humidity (RH) (> 90%), oxygen availability (ISS’ atmosphere), warm temperature (25 °C), and darkness, rendering the ideal environment for fungal growth. To activate the samples, the PHABs were opened, the plastic bag around the four plates was removed, and the two pieces of absorbent mat (one on the lid and one on the floor of the PHAB) were wet with three water syringes (9 mL total of water per PHAB). Then, the plates were returned to the PHAB, and the PHAB was placed at 25 °C undisturbed for 10 d 2 h, 14 d 19 h, or 20 d 3.5 h.

After the designated incubation period, the samples were “terminated” while in microgravity to fix the spaceflight morphology or preserve the spaceflight transcriptome. Termination of the experiment consisted of flooding the wells with either 4% PFA (morphology samples) or RNAlater (transcriptomic samples), and took place inside the Life Sciences Glovebox (LSG). To terminate the samples, the Breathe-Easy membrane was pierced with a needle to inject enough fixative or preservative into each well. The punctured membranes were dried from the outside with sterile gauze, and sterile RNase-free sealing strips (Sigma-Aldrich, Cat. Z707392) were placed over to seal the holes left by the needles. Terminated plates fixed with 4% PFA were stowed at 4 °C, and plates preserved with RNAlater at −95 °C on orbit; immediately upon receiving them back on Earth, they were stowed at 4 and −80 °C, respectively, to prevent sample degradation.

### 2.4. Post-Flight Biofilm Morphology Data Acquisition

Upon return to Earth, the plates with the fixed coupons were maintained at 4 °C. When ready for morphology data acquisition, the coupons were transferred (without touching the coupon’s surface with the biofilm) into a new 24-well plate in batches of six or eight (space and Earth control of a respective coupon treated on the same session). Each coupon was stained with a mixture of 400 µL of Calcofluor White (Sigma-Aldrich, 18909-100ML-F), staining the chitin present in fungal cell walls, and 400 µL of Biofilm Ruby (Invitrogen, Waltham, MA, USA, F10318), marking proteins present in the extracellular matrix, for 30 min protected from light. Then, each coupon was gently dipped in distilled water four times to remove excess stain and glued onto a microscope slide. To prepare the slides for microscopy, 65 µL of VectaShield anti-fade hardening mounting medium (Vector Laboratories, Burlingame, CA, USA, H-1900) and a cover slip were gently placed on top of the coupon. The coupons then stayed protected from light for 2.5 h to allow the mounting medium to harden. 

### 2.5. Image Acquisition

Microscopy images were acquired with a Nikon A1 Confocal Microscope using the 40 × 0.6NA objective and the NIS Elements 5.21.03 (Nikon Instruments Inc., Tokyo, Japan) imaging software. Two emission filters were used: 425–475 nm for Calcofluor White and 500–550 nm for Biofilm Ruby, with a 405 nm and 561 nm laser, respectively. The 3D structure of the biofilm was captured using a z-stack (step size of 0.243–2.096, 30 slices minimum), and each coupon was imaged in one large field (2 × 2 fields of 302.35 µm × 302.35 µm each, for a total of 604.70 µm × 604.70 µm). The coupons were always imaged in the center to avoid potential user bias looking for a particular section of biofilm.

### 2.6. Qualitative Analysis

After imaging the biofilms, a qualitative analysis was performed on all the sample images using the NIS Elements Viewer 5.21.00 software (Nikon Instruments Inc., Tokyo, Japan). The qualitative analysis consisted of describing the morphology of the biofilms to find differences and similarities of the biofilms, namely observations regarding structure and distribution, based on material surface, incubation time, and gravity condition.

### 2.7. Conversion from .nd2 Files to .tiff Files and Slicing of Images

The images taken with the microscope were stored as .nd2 files, which is an incompatible format for quantitative analysis with COMSTAT2. Thus, all the images were converted to .tiff files using ImageJ v1.48 [34]. To this end, the nd2 file was opened in the ImageJ software and the two channels were merged, and then the composite was subsequently converted to RGB color, 8-bit, and finally .tiff file. The ImageJ plugin Slice Remover [35] was used to remove the slices of the image stacks that did not have any relevant biofilm information (i.e., underneath the bottom layer and over the topmost layer where no biofilm signal was recorded). This ensured the software would not yield incorrect biofilm data.

### 2.8. Quantitative Analysis and Quality Control

For the analysis process, COMSTAT 2 software v2.1 [36,37,38] was used as a plugin in ImageJ. The images were converted into OME-Tiff files using the ImageJ macro “Convert to OME-Tiff” to ensure compatibility with COMSTAT 2. Each OME-Tiff file was analyzed using the following parameters: (i) automatic thresholding with Otsu’s Method, and (ii) no connected volume filtering. The measurements produced by the software were (a) biomass and (b) thickness distribution. A proxy for biomass, which will be referred to as biomass for the purpose of this investigation, is calculated in COMSTAT2 as the ratio of volume per unit area. Using the data for 0 µm thickness, we calculated the (c) surface area coverage of the biofilm. As a quality control step, each mask created by COMSTAT2 was compared with the original .nd2 files to check for resemblance between the two files. The images which were far off from the original image were re-analyzed with a manual threshold instead of automatic threshold.

The biomass of each of the *t_0_* samples was produced by COMSTAT2 using the same methodology. Thickness and surface area were not calculated as these samples only contained spores which could be stacked or clumped together, potentially leading to a false statistical significance when comparing amount of spores. Biomass was hence deemed the best parameter to examine for this set of samples as it takes into account the overall volume occupied by the spores.

### 2.9. Statistical Analysis

Statistical analysis was performed using R version 4.1.2 in RStudio (v4.1.2; R Core Team 2021). The data obtained from the quantitative analysis were first tested for normality and homogeneity to determine if parametric or nonparametric statistical tests would be used. Normality was tested using the Shapiro–Wilk test and homogeneity was tested using Levene’s test. The data did not comply with the assumptions of normality and homogeneity (only a few subsets complied); therefore, nonparametric statistical tests were used. The Kruskal–Wallis test and Dunn’s test with Bonferroni correction were used to compare the median values of biomass and surface area coverage of biofilms based on the gravity and incubation time per material as well as to compare the differences between materials. A significance level of 0.05 was used (*p* < 0.05). 

## 3. Results

The results presented below in Section 3.3, Section 3.4, Section 3.5, Section 3.6, Section 3.7, Section 3.8 and Section 3.9 are all statistically significant results. Any comparisons that were not statistically significant are not reported. All figures with statistically significant results are indicated using brackets. Comparisons without brackets indicate a lack of statistically significant differences.

### 3.1. RH and Temperature Profiles

The temperature and RH registered inside the PHABs (Figure 5) shows a successful increase in temperature and humidity to promote fungal growth. The average temperature of incubation (activation to termination) was 24.8 ± 2.4 °C and 25.6 ± 0.9 °C for ground and spaceflight samples, respectively. The RH during incubation was 96.6 ± 6.0% for all spaceflight samples, while for Earth PHABs, the humidity was less stable for some cases. The average humidity on the ground was 56.1 ± 9.6%, 91.9 ± 5.9%, and 48.4 ± 16.9% for 10, 15, and 20 days of incubation, respectively.

### 3.2. t_0_ Samples

As expected, low biofilm biomass was observed on the *t*_0_ samples for each material (Table 2). No significant differences in mass were found between materials at *t*_0_.

### 3.3. Carbon Fiber

No significant differences in *P. rubens* biofilm mass and thickness were observed when comparing incubation time and gravity conditions on carbon fiber. Nevertheless, there were two statistical differences in surface area coverage between gravity conditions. When grown for 10 days, biofilms formed in space had 22.7% less surface area coverage compared to samples grown on Earth (*p* < 0.05). However, after 15 days of growth, the spaceflight samples had 37.8% more surface area coverage than the Earth controls (*p* < 0.05) (Figure 6).

### 3.4. Stainless Steel 316

There were no significant differences observed in *P. rubens* biofilm mass or thickness when compared throughout the gravity conditions and incubation times. Nevertheless, there were two statistical differences in surface area coverage under two comparisons. Biofilms formed on Earth showed differences between days 10 and 20, where the surface area coverage had an 85.8% decrease from day 10 to day 20 (*p* < 0.05), as seen in Figure 7a. Comparing between spaceflight and Earth, the only difference observed was on day 15, when the samples grown in space had a 30.7% increase in surface area coverage with respect to the 1 g controls (*p* < 0.05); such an increase in microgravity was also noticeable in the microscopy images (Figure 7b,c).

### 3.5. Quartz

There were no significant differences observed in mass or surface area coverage when comparing incubation times and gravity conditions of *P. rubens* biofilms grown on quartz. Nevertheless, biofilm thickness after 20 days had an 88.5% decrease compared to 15 days of incubation time in space conditions (*p* < 0.05). When comparing the samples grown for 20 days, those grown in space had 58.8% less thickness compared to Earth samples (*p* < 0.05), shown in Figure 8.

### 3.6. Titanium Alloy

The biofilm mass (Figure 9a) and thickness (Figure 9c), when grown for 20 days on Earth, showed a significant decrease compared to samples grown for 15 days with an 8% decrease in mass and a 52.8% decrease in thickness (*p* < 0.05). Such a biofilm reduction (mass and thickness) was not observed in microgravity between 15 and 20 days. Surface area coverage (Figure 9b) of samples grown for 10 days on Earth was 4.6 times higher than those grown for 20 days under the same conditions (*p* < 0.05). Significant changes between space and Earth samples were also observed in biofilm thickness. After 10 days of incubation, samples grown in space were 2.3 times thicker compared to samples on Earth (*p* < 0.05). A similar observation was seen in the samples after 20 days, with samples grown in space being 1.97 times thicker compared to samples on Earth (*p* < 0.05) (Figure 9c).

### 3.7. Silicone, MIT Nanograss, and Aluminum Alloy

Biofilms grown on Silicone, Nanograss, and Aluminum Alloy presented no significant differences when compared by gravity and incubation time regarding biofilm mass, thickness, or surface area coverage.

### 3.8. Effect of Surface Material on Earth

After 15 days growing in Earth conditions (Figure 10), the biofilm surface area on silicone was 13% higher than the biofilm surface area on the Nanograss material (*p* < 0.05). There were no significant differences in biofilm thickness and mass between materials. There were also no significant differences between materials in biofilm thickness, mass, and surface area for Earth samples grown for 10 and 20 days.

### 3.9. Effect of Surface Material in Space

#### 3.9.1. Space Samples after 10 Days

After growing in space conditions for 10 days, Stainless Steel had 19 times more biofilm mass (*p* < 0.05) (Figure 11a) and 9% more surface area coverage (*p* < 0.01) (Figure 11b) than Nanograss. There was no significant difference in biofilm thickness between materials grown for 10 days in space.

#### 3.9.2. Space Samples after 15 Days

Biofilm mass was significantly different between Nanograss and four other materials when grown in space for 15 days. Compared to the biofilm mass on Nanograss, 27-, 32-, and 30-fold increases in mass were observed on Stainless Steel, Quartz, and Titanium Alloy, respectively (*p* < 0.05) (Figure 12a). Stainless Steel and Titanium Alloy not only had more biofilm mass than Nanograss, but also had 5% (*p* < 0.01) and 4% (*p* < 0.05) more biofilm surface area coverage, respectively (Figure 12b). There were no significant differences in biofilm thickness when grown in space conditions for 15 days.

#### 3.9.3. Space Samples after 20 Days

After growing for 20 days in space, Aluminum Alloy’s biofilm was 25% thicker than Nanograss’s biofilm (*p* < 0.05) (Figure 13). There were no significant differences in biofilm mass or surface area after 20 days of growth in space conditions.

## 4. Discussion

The primary aim of this investigation was to characterize *P. rubens* biofilm morphology on seven clinically or spaceflight relevant materials both in space and on Earth. All materials were susceptible to fungal biofilm formation, some more than others, regardless of the gravitational condition. The fungal samples had equivalent initial amounts of spores on all material substrates, as there were no significant differences in biomass after 6 h of incubation. Although the temperature was kept close to 25 °C during the incubation time across samples, and despite our efforts to provide a stable environment, the humidity presented significant variations. Specifically, the samples incubated for 10 and 20 days on Earth had RH of approximately half that of the rest of the conditions. Since each PHAB contained samples of all material coupons, this does not affect the comparisons across materials, but it could interfere with comparisons across incubation times and between gravitational regimes. 

### 4.1. The Effect of Microgravity on Biofilm Growth

One of the primary dependent variables examined was fungal biofilm formation in microgravity compared to Earth’s gravity. Three cases of significantly increased biofilm growth in microgravity were observed: Carbon Fiber at day 15 (surface area increase), Stainless Steel at day 15 (surface area increase), and Titanium Alloy at day 10 and day 20 (thickness increase). However, there were two cases that showed the contrary with significantly decreased biofilm growth in microgravity compared to Earth: Carbon Fiber at day 10 (surface area decrease) and Quartz at day 20 (thickness decrease). These two cases, where fungal biofilms on Earth grew better than in space, are of particular interest because those samples correspond to the conditions (Earth at 10 days and 20 days) that experienced fluctuations of humidity (Figure 5). The ideal RH for *P. chrysogenum* is 90%, and decreases in fungal growth below this RH have been observed [39]. Previous studies have shown that drops in humidity can immediately stop growth and cause the dehydration of *P. rubens* [40]. On occasion, the fungus is able to start growing again when the humidity increases, but other times, RH fluctuations (of as little as 3 h) can suppress growth permanently, so only new germination can occur [40]. Additionally, there is evidence that after being exposed to high humidity, *P. rubens* cannot restart growth following a desiccation event of RH below 75% [41]. Since the humidity fluctuations in Earth PHABs of 10 and 20 days lasted for days and the PHABs reached RH of 50% and below, we expected the fungal growth to be impaired for those samples. Hence, it is possible that if RH values would have been equivalent between flight and ground in the day 10 and 20 PHABs, even more biofilm growth could have been observed on Earth. Nevertheless, this would not change our relative observation presented here—likely except for Titanium Alloy at 20 days, results that may be an artifact of the RH challenges. Quartz at 20 days was an exception to this trend as an increase in biofilm formation was seen despite lower RH values. While we have no clear explanation as to why this occurred, a previous study classified *P. chrysogenum* (the previous species name for *P. rubens*) as xerotolerant, meaning it can undergo a complete life cycle at low RH values and is considered resistant to dry environments [42]. While this was not studied specifically in this investigation, how *P. rubens* survives in dry environments should be further examined.

With that in mind, our results suggest mixed effects of microgravity on biofilm growth dependent on the material surface and without a clear trend in time. It is important to note the significant outlier in the Carbon Fiber 10-day sample grown on Earth and the effect of removing this outlier from the statistical test. When removed, the comparison between Earth and space sample groups is no longer significant; however, all of the other observed differences remain the same, which implies mixed effects of microgravity on biofilm growth. Despite the changes in biofilm growth in microgravity, no consistent visual trends in morphological shape (overall 3D structure) across gravity conditions were observed. Therefore, in our experiment, microgravity did not impact the fungal shape of the biofilm. The mechanisms behind the changes in fungal biofilm formation in microgravity are still unclear and need further investigation. One potential mechanism behind the changes observed in this investigation could lie in how different fungi sense gravity. The octahedral crystal matrix protein (OCTIN) is the structural protein comprising the vacuolar protein crystals that the fungal family Mucorales uses to sense gravity [43]. While this protein does not exist in Penicillium species, other unknown gravity sensing proteins could be present. Research into how other fungal families, such as Penicillium, might be capable of sensing gravity is a crucial next step to understanding fungal biofilm growth in microgravity and the changes in biofilm formation observed in this study.

### 4.2. The Effect of Material Surface on Biofilm Growth

After qualitative analysis of the fungal biofilms, no difference in biofilm shape was observed across materials (Appendix B). However, significant differences in mass, thickness, and surface area coverage of the biofilms between materials were identified.

The Nanograss material was of particular interest in this investigation as a potential biofilm-resistant material. Nanograss was the most efficient material at reducing fungal biofilm formation, both in space and on Earth. One potential explanation for the reduced biofilm formation is a reduced strength of adhesion. Spore adhesion was the same across all materials by 6 h time (Appendix A), meaning all spores were at minimum equivalently loosely adhered to the material surface. Adhesion strength, how strongly the fungi adheres to the material, and how much force the spores or biofilm can withstand before being removed are dependent on various material characteristics such as hydrophobicity, roughness, and charge [44]. Hydrophobicity is unlikely to be a factor contributing to the reduced biofilm growth. This is due to hydrophobins, which are proteins that help to determine a fungal spores’ hydrophobicity. One hydrophobin, RodA, has been shown to coat the outside of spores, forming a hydrophobic rodlet layer in *Aspergillus fumigatus*, and other hydrophobins have been identified in hyphae [45,46,47]. An ortholog RodA gene with 100% query cover (E value 0.0, and 90.40% identity) was found in *P. rubens* (Accession No. AM920436.1) using NCBI’s BLAST Tool, meaning that *P. rubens* could also have hydrophobic spores if this gene is being expressed. Nanograss is a highly hydrophobic material and therefore should allow the *P. ruben* spores to strongly adhere. Therefore, hydrophobicity is not the factor interfering with spore and hyphal adhesion preventing biofilm formation on Nanograss.

Other factors such as surface roughness and charge are likely at play in preventing biofilm growth on Nanograss [44]. In the majority of previous research, *P. rubens* biofilms have been grown on gypsum, and data on the effect of roughness and surface charge on *P. rubens* biofilm formation are not available. Additionally, data on the roughness and charge of the materials used in this investigation are not available as these properties are manufacturer- and sometimes batch-dependent. Future experiments should measure this material’s properties to examine the effect of different material charges and roughness on biofilm growth on hydrophobic materials. Due to the decreased biofilm formation on Nanograss, this material warrants further studies with varying conditions and strains to determine its full potential as an anti-microbial material.

### 4.3. The Effect of Incubation Time on Biofilm Growth

A decrease in biofilm growth was observed as incubation time increased in three cases: Titanium Alloy Earth samples (mass and thickness decreased between day 15 and day 20 and surface area decreased between day 10 and day 20), Quartz space samples (thickness decrease between day 15 and day 20), and Stainless Steel Earth samples (surface area decrease between day 10 and day 20). Since there was only a thin coat of PDB on the surface of the coupons, it is possible the nutrients were depleted as incubation time increased, causing the fungal biofilms to be in a starvation state. When in unfavorable conditions, small groups of fungal persister cells will form and remain even as other fungal cells die [48]. These cell populations have decreased metabolism and growth, allowing the biofilm to remain at a stable population level until conditions improve and growth can resume [49]. Furthermore, *Candida albicans*, another fungal species known for biofilm formation, can maintain a consistent population in starvation after an initial drop in cell population [50]. If the biofilms have depleted resources by day 15, a decrease in overall biofilm mass, thickness, and surface area can be expected as the cell population will start to die but maintains a minimum population with the assistance of persister cells. The reduced biofilms observed at day 20 could contain these persister cells, which would allow the biofilm to continue growth during a longer incubation time if the environment becomes favorable again, which could present a possible risk for future long-term missions. Further studies should be conducted to observe the effect of adding more nutrients to the fungal population after 20 days in microgravity. In the future, analyses of persister cells within the biofilm and their gene expression should be conducted, comparing the number of persister cells formed both in space and on Earth. This is especially relevant because the persister cells are more resistant to antimicrobials, and in confined spaces such as spacecraft, they could pose an increased health risk for astronauts.

Another possibility for the decrease in biofilm formation on day 20 due to depleted nutrients is the weakening of cell–cell and cell–surface adhesions. Fungi have two types of adhesion proteins which are vital for cell–cell adhesion and cell–surface adhesion, lectin-like adhesions and sugar-independent adhesins [51]. Depleted nutrients decrease the amount of available sugars for the adhesins to bind to, potentially decreasing the strength of cell–surface adhesion. Whether the sugar-independent adhesions are sufficient to maintain a strong attachment is unknown and was not studied in this investigation. However, if the sugar-independent adhesions are insufficient, then fungi with a loose attachment were potentially washed away in the gentle washing steps during staining. Further studies should be conducted to determine if the depleted resources decreased the strength of biofilm adherence.

Altogether, the results show that microgravity does not have a strong effect on biofilm shape and morphology of *P. rubens* up to 20 days of incubation. Even in the few cases where there was increased biofilm formation in microgravity, this effect was not maintained in time. Additionally, due to the reduced fungal biofilm formation, both in space and on Earth, Nanograss could be considered as a potential anti-microbial material for certain spacecraft equipment. The results presented in this manuscript can contribute to the efforts of fungal biofilm control in space, but long-term tests (>20 days) as well as complementary transcriptomic and proteomic analyses are highly recommended, as microgravity effects were both material- and incubation-time-dependent.

## Figures and Tables

**Figure 1 life-13-01001-f001:**
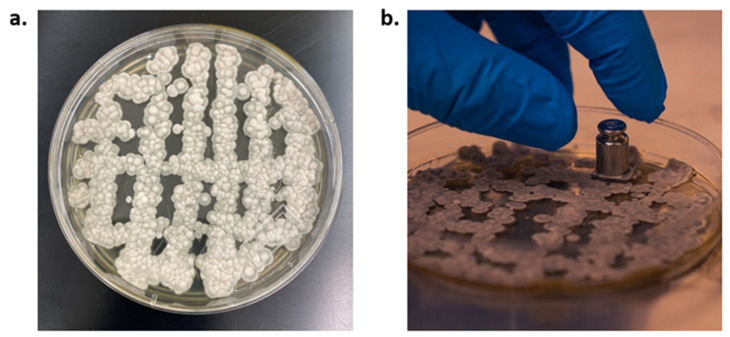
Inoculation of spores of *P. rubens* on material coupon surfaces. (**a**) Example of 6 × 6 grid of *P. rubens* 6-day-old mold on PGA plates, and (**b**) inoculation of mold spores with 5 g weight.

**Figure 2 life-13-01001-f002:**
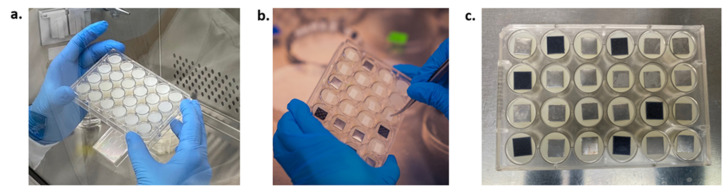
Twenty-four-well plates with (**a**) silicone risers only, (**b**) placement of inoculated coupons over silicone risers, and with (**c**) silicone risers, samples, and membrane film.

**Figure 3 life-13-01001-f003:**
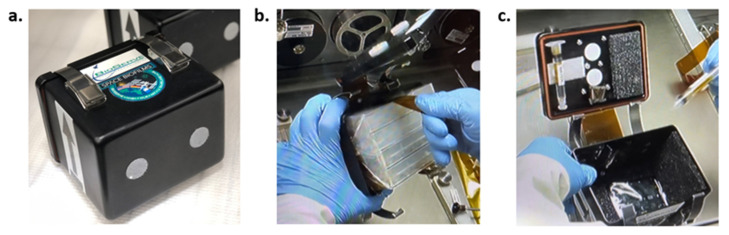
BioServe Space Technologies’ PHAB in Space Biofilms configuration. (**a**) Loaded PHAB. (**b**) Opened PHAB with four 24-well plates (stacked and inside a bag) being removed. (**c**) Inside view of the PHAB with water syringes, adsorbent mats, and HOBO in the bottom. Photo credit: BioServe and NASA.

**Figure 4 life-13-01001-f004:**
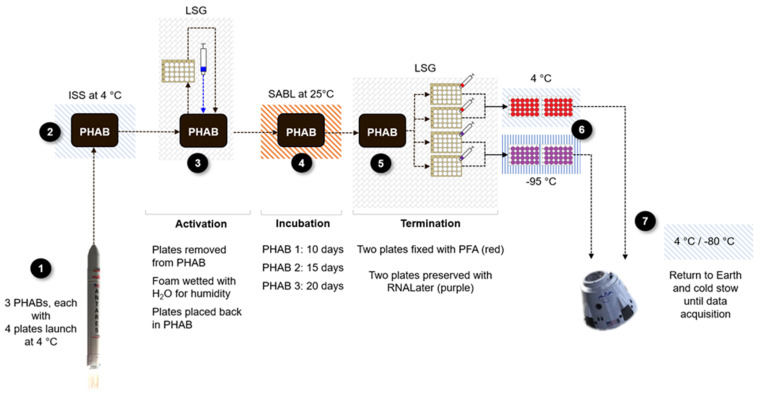
Concept of operations for the Fungal Space Biofilms experiment. LSG = Life Sciences Glovebox, SABL = BioServe’s Space Automated Bioproduct Laboratory incubator on board ISS.

**Figure 5 life-13-01001-f005:**
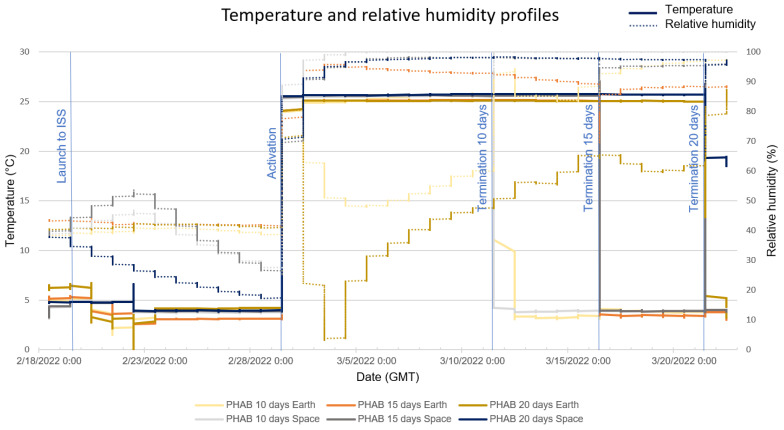
Temperature and RH profiles of the experiment. Temperature data in solid lines and RH in dotted lines. Relevant experiment activities labeled in blue.

**Figure 6 life-13-01001-f006:**
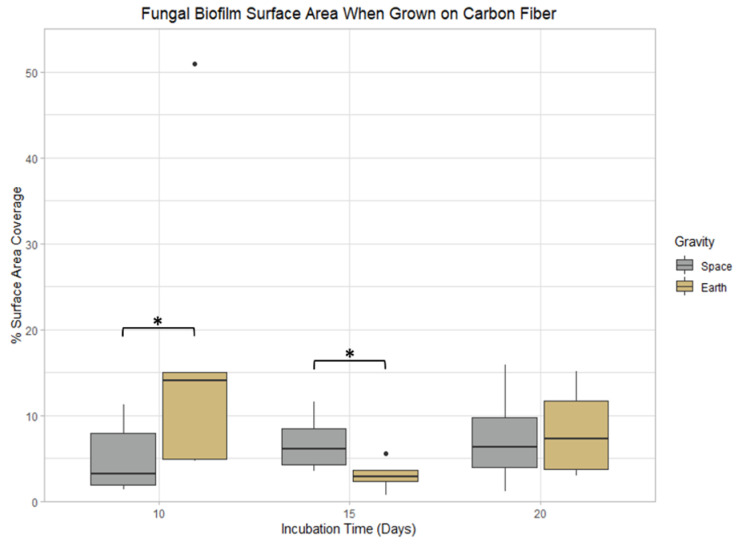
Incubation time and gravity condition comparisons of biofilm surface area coverage when grown on carbon fiber. *: 0.01 < *p* ≤ 0.05.

**Figure 7 life-13-01001-f007:**
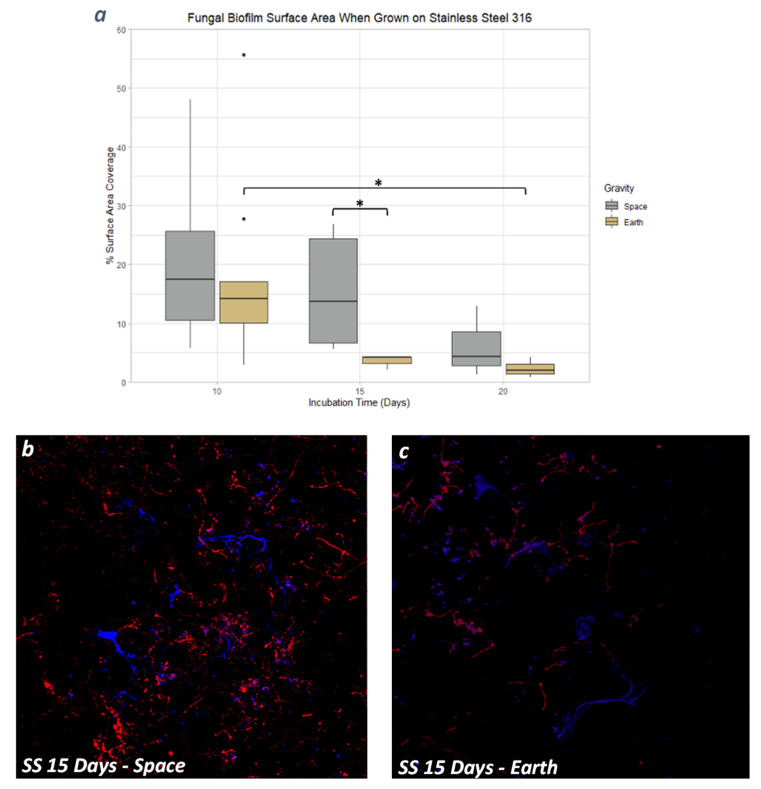
(**a**) Fungal biofilm surface area coverage on Stainless Steel 316. Images of fungal biofilm after 15 days of incubation when grown in space (**b**) and on Earth (**c**), showing differences in surface area coverage. The Calcofluor White stain in blue binds to cellulose and chitin. The Biofilm Ruby stain in red binds to components of the biofilm’s extracellular matrix. *: 0.01 < *p* ≤ 0.05.

**Figure 8 life-13-01001-f008:**
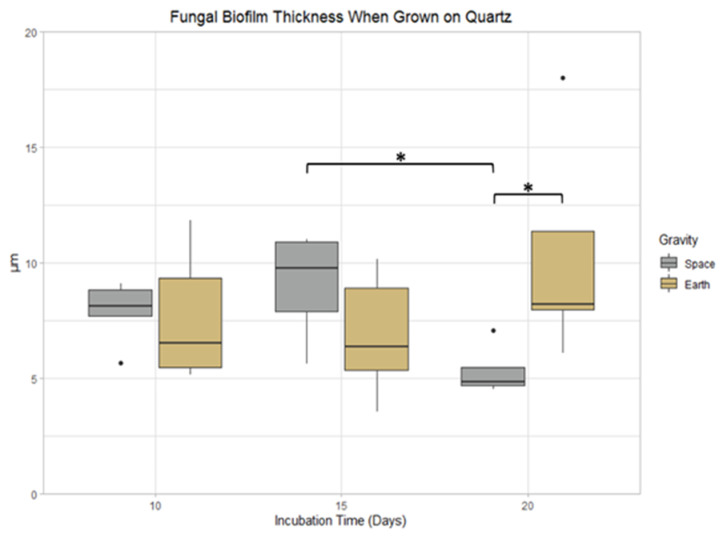
Incubation time and gravity condition comparisons of biofilm thickness when grown on quartz. *: 0.01 < *p* ≤ 0.05.

**Figure 9 life-13-01001-f009:**
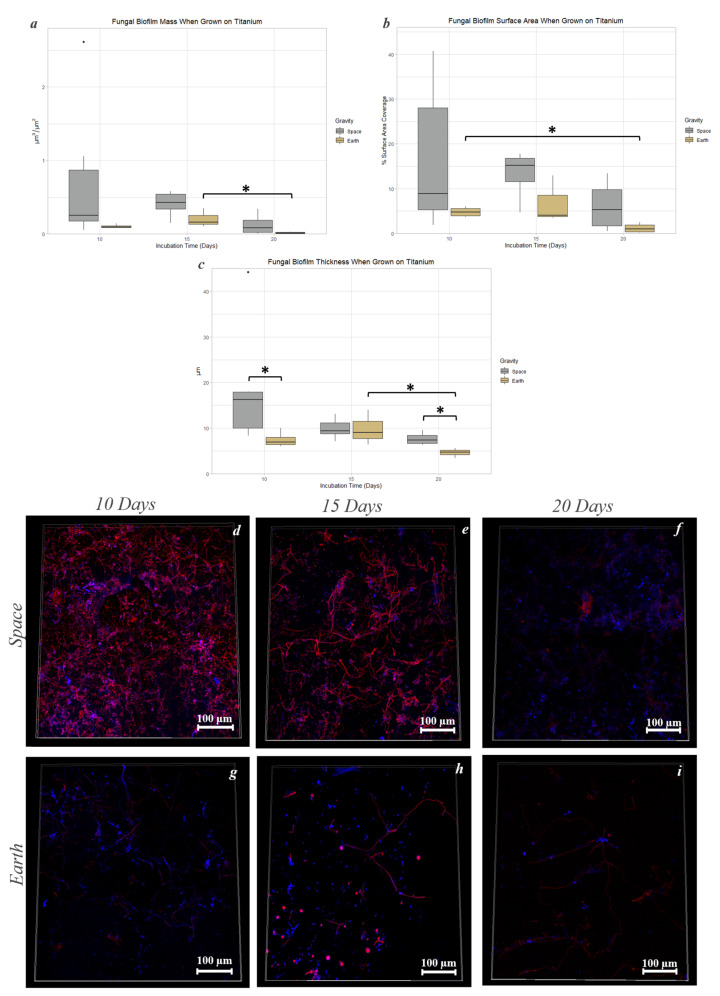
Incubation time and gravity condition comparisons of biofilm mass, surface area, and thickness when grown on Titanium Alloy. Graphs show fungal biofilm mass (**a**), surface area (**b**), and thickness (**c**) in space and Earth conditions after 10, 15, and 20 days. Images of both space and Earth samples after 10, 15, and 20 days demonstrate fungal biofilm differences at 40× (**d**–**i**). The Calcofluor White stain in blue binds to cellulose and chitin. The Biofilm Ruby stain in red binds to components of the biofilm’s extracellular matrix. *: 0.01 < *p* ≤ 0.05.

**Figure 10 life-13-01001-f010:**
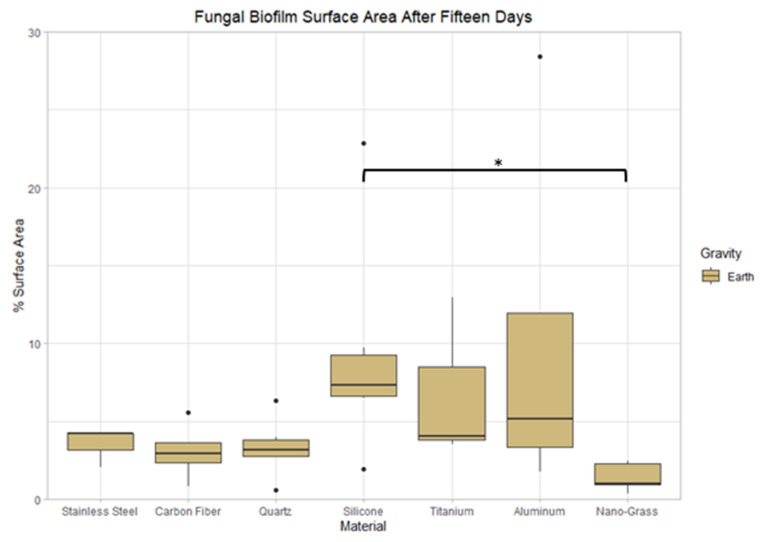
Comparison of biofilm surface area after 15 days of incubation in Earth conditions on seven different materials. The nomenclature for the significant values is as follows: *: 0.01 < *p* < 0.05.

**Figure 11 life-13-01001-f011:**
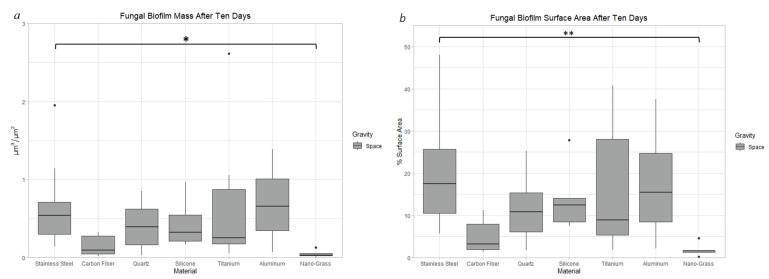
Comparison of biofilm mass (**a**) and surface area (**b**) after 10 days of incubation in space conditions on seven different materials. The nomenclature for the significant values is as follows: *: 0.01 < *p* < 0.05, **: 0.001 < *p* < 0.01.

**Figure 12 life-13-01001-f012:**
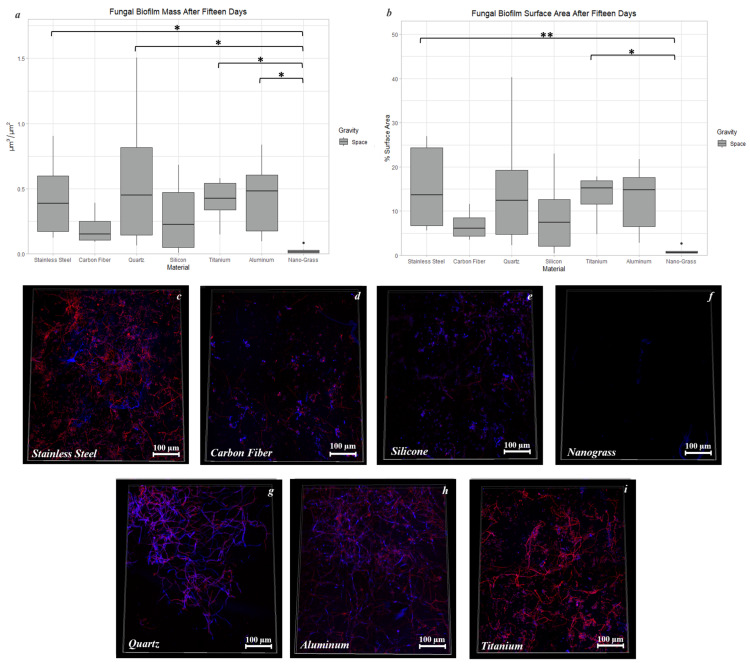
Comparison of biofilm mass (**a**) and surface area (**b**) after 15 days of incubation in space conditions on seven different materials. Images of fungal biofilms grown on Stainless Steel (**c**), Carbon Fiber (**d**), Silicone (**e**), Nanograss (**f**), Quartz (**g**), Aluminum Alloy (**h**), and Titanium Alloy (**i**) after 15 days of incubation time in space conditions. The Calcofluor White stain in blue binds to cellulose and chitin. The Biofilm Ruby stain in red binds to components of the biofilm’s extracellular matrix. The nomenclature for the significant values is as follows: *: 0.01 < *p* < 0.05, **: 0.001 < *p* < 0.01.

**Figure 13 life-13-01001-f013:**
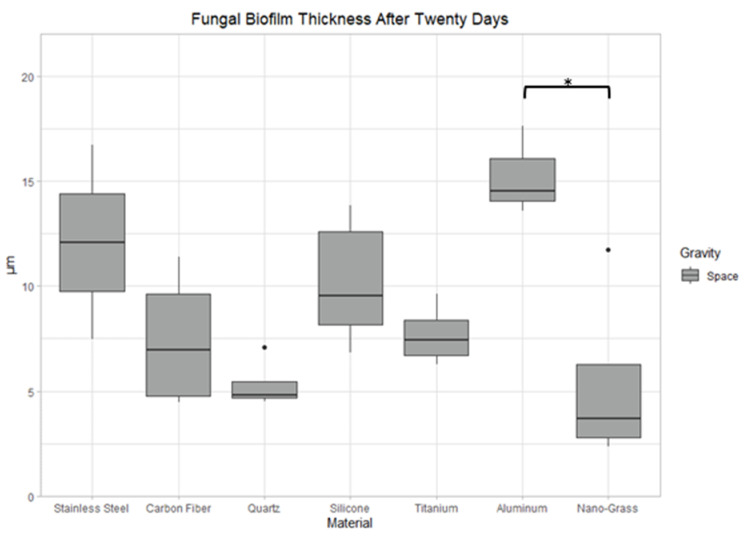
Comparison of biofilm thickness after 20 days of incubation in space conditions on seven different materials. The nomenclature for the significant values is as follows: *: 0.01 < *p* < 0.05.

**Table 1 life-13-01001-t001:** Material coupons and cleaning methods used in the experiment. Amount accounts for space, Earth, and *t*_0_ sets.

Material Coupon	BioServe Part No.	Amount	Cleaning Method
Aluminum Alloy Al6061	4999 MPPR-311-4	84	Ultrasonic
Stainless Steel 316	5183 MPPR-311-6	120	Ultrasonic
Quartz	6090 MPPR-311-8	84	Ultrasonic
Silicone	6092 MPPR-311-2	84	Bag washed
Titanium Alloy Ti-6Al-4V	4999 MPPR-311-3	84	Ultrasonic
MIT Nanograss	2022	78	Not washed, prepared in sterile conditions
Carbon Fiber	5012 MPPR-311-7	84	Ultrasonic

**Table 2 life-13-01001-t002:** Median values of fungal biofilm mass on *t_0_* per material. Interquartile range Q3–Q1 (IQR) is indicated in parenthesis.

Material	Biomass (µm^3^/µm^2^)
Stainless Steel 316	0.0267 (0.0142)
Carbon Fiber	0.0205 (0.0155)
Silicone	0.0773 (0.0375)
Quartz	0.0133 (0.0074)
Titanium Alloy	0.0112 (0.0449)
Aluminum Alloy	0.0052 (0.0061)
Nanograss	0.0096 (0.0162)

## Data Availability

All microscopy files and raw data obtained from COMSTAT2 can be found in NASA’s Physical Sciences Informatics (PSI) data repository (https://psi.nasa.gov). Project ID TBD.

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
