# Peer review of "Morphology of Penicillium rubens Biofilms Formed in Space"

_life, 2023, doi:10.3390/life13041001_

Round 1
Reviewer 1 Report
The manuscript is well-written and generally the results are satisfactorily provided and discussion is sufficient. However, I do think the manuscript could be improved by including and discussing previous biofilms-in-space research, particularly Kim et al. 2013 (PLoS ONE). The results from previous studies could be compared to the results in this manuscript, and more discussion could be provided regarding minimizing fungal contamination in future space stations/missions and what materials might work best.
Line items:
Simple summary: the sentence at lines 20-21 seems to contradict the line before it.
Introduction:
Line 59-60: Need references for past studies.
Line 64: for consistency: sometimes the term ‘fungal biofilm’ is used, and other times it is ‘fungi biofilm’.
Line 110: I think it could be argued that “little is known about microbial behavior when grown as biofilms”. I also assume this is meant “in space.” Such as: Kim et al. 2013, Flores et al. 2022 (ref #25), Cotiin and Rettberg 2019, Zhao et al. 2019, Marra et al. 2023, Loudon et al. 2018, etc.
Line 111-114: What about Kim et al. 2013? I wouldn’t dismiss previous and ongoing biofilms research.
Line 119-120: Why aren’t these previously-published studies referenced in line 110 then?
Line 127: References needed.
Line 147: Is there any reference for MIT nanograss? It is not mentioned in reference #7.
Line 153: What is LIS?
Line 214: Reference error.
Line 343: Why are these references not numbered?
Line 348: how does the program measure biomass? Wouldn’t the program be measuring thickness and surface area coverage to determine biomass, if a typical cell size/weight was given?
Line 385: Table 2: what are the values in parentheses? Error? Were t0 samples only measured for biomass? Or were samples too limited to be measured for biofilm thickness and surface area coverage? If biomass is given in um^3/um^2, isn’t this the same as thickness and surface area coverage, and not actually a different parameter?
Line 388-: For Sections 3.3 through 3.9.3, are all comparisons (whether significantly different or not significantly different) mentioned? For example, in Figures= 11, was nanograss significantly different from all other materials, or just stainless steel?
Line 395: Figure 6: What do the results indicate if that significant outlier (for the Earth sample with 50% coverage) is removed?
Line 424: Figure 8: The comparison between 20 days in space and 10 days in space is not noted on this figure (lines 420-421).
Line 435: Here you give the actual p-value, whereas everywhere else it is simply written as p<0.05.
Line 455: This sentence says 15 days whereas Figure 10 figure caption says 10 days.
Line 470: Figure 11a: I think the bracket is meant to go between stainless steel and nanograss?
Line 505: Could a difference have been seen if they were incubated for longer? 12 hours? 24 hours?
Line 508-511: Could another set of samples be run under Earth conditions at a higher relative humidity, just for further comparison?
Line 517: What is meant here by “growth”? Biomass, biofilm thickness or surface area coverage?
Line 538: what is meant by shape? Up until now, only biofilm biomass, biofilm thickness, and surface area coverage have been mentioned.
Line 553: This seems pretty subjective, as there appears to be differences between space and Earth fungal biofilms from the images (though difficult to see) provided in Appendix B. For example, simply the amount of cellulose/chitin vs. ECM proteins in some of the images.
Line 557-559: How is adhesion strength a possible explanation when adhesion was similar across all materials?
Line 582: what is meant by growth?
Line 909: Is reference 27 a preprint? That is all I can find. It also lists Zea as the first author.
There are inconsistencies and missing information amongst the references list. Examples: Ref #9, Ref #14, Ref #19, etc.
Reviewer 2 Report
The authors investigated the morphology of Penicillium rubens biofilms grown on seven different materials on board the ISS for 10, 15 and 20 days. The tests were performed in parallel with their corresponding ground controls. The assessment is unprecedented because it considers that fungal growth in space has not yet been evaluated and compared to terrestrial controls. In addition, the growth of fungal biofilm on a new material, nanograss, was evaluated. Due to constant fungal contamination in spaceflight equipment and habitats associated with health issues, the article is pertinent and relevant to the area.
Some questions:
Line 13: The sentence begins with mold (fungal biofilm). In this context, is all mold considered a fungal biofilm?
Line 48: Current estimates suggest that up to 80% of bacterial and archaeal cells reside in biofilms and, not “all microorganisms”. Update the reference (it's over 20 years old). Suggestion: https://doi.org/10.1038/s41522-021-00251-2
Line 65: fungal biofilms, or molds…is correct?
Lines 85 – 87: What would be the possible reasons for this increase in aggressiveness acquired by these post-flight lineages? Are there current articles that could justify this effect?
Line 523: – “Previous studies have shown that drops in humidity can immediately stop
growth and cause the dehydration of P. rubens”, however, there is no mention of the optimal RH for growth of this fungus in the text.
Line 568: “Therefore, hydrophobicity is most likely not interfering with spore adhesion but potentially on hyphal adhesion on Nanograss”. Rodlet structures (Class I hydrophobins) can be found in aerial structures such as hyphae and conidia. If hyphae have rodlets would these also be hydrophobic, so could they bind to nanograss? To see: https://doi.org/10.3390/microorganisms10081498
It is observed that PHAB 20 days on Earth (Figure 5) had a sharp decrease in relative humidity between the 2nd and 3rd of May, with values below 10%. The text mentions studies where values lower than 75% RH could interfere with the re-establishment of growth, and this really happened with Titanium. But what explains the accentuated growth of the fungus in Quartz, on Earth, in 20 days, even with a decrease in RH?
